# Sorting out Lipschitz Function Approximation

## Abstract

Training neural networks subject to a Lipschitz constraint is useful for generalization bounds, provable adversarial robustness, interpretable gradients, and Wasserstein distance estimation. By the composition property of Lipschitz functions, it suffices to ensure that each individual affine transformation or nonlinear activation function is 1-Lipschitz. The challenge is to do this while maintaining the expressive power. We identify a necessary property for such an architecture: each of the layers must preserve the gradient norm during backpropagation. Based on this, we propose to combine a gradient norm preserving activation function, GroupSort, with norm-constrained weight matrices. We show that norm-constrained GroupSort architectures are universal Lipschitz function approximators. Empirically, we show that norm-constrained GroupSort networks achieve tighter estimates of Wasserstein distance than their ReLU counterparts and can achieve provable adversarial robustness guarantees with little cost to accuracy.

Constraining the Lipschitz constant of a neural network ensures that a small change to the input can produce only a small change to the output. For classification, a small Lipschitz constant leads to better generalization (Sokolić et al., 2017), improved adversarial robustness (Cisse et al., 2017; Tsuzuku et al., 2018), and greater interpretability (Tsipras et al., 2018). Additionally, the Wasserstein distance between two probability distributions can be expressed as a maximization problem over Lipschitz functions (Peyré & Cuturi, 2018). But despite the wide-ranging applications, the question of how to approximate the class of Lipschitz functions with neural networks remains largely unanswered.

Existing approaches to enforce Lipschitz constraints broadly fall into two categories: regularization and architectural constraints. Regularization approaches such as double backprop (Drucker & Le Cun, 1992) or the gradient penalty (Gulrajani et al., 2017) perform well in practice, but do not provably enforce the Lipschitz constraint globally. On the other hand, norm-constrained architectures place limitations on the operator norm (such as the matrix spectral norm) of each layer's weight matrix (Cisse et al., 2017; Yoshida & Miyato, 2017). These techniques provably satisfy the Lipschitz constraint, but this comes at a cost in expressive power. E.g., norm-constrained ReLU networks are provably unable to approximate simple functions such as absolute value (Huster et al., 2018).

We first identify a simple property that expressive norm-constrained 1-Lipschitz architectures must satisfy: gradient norm preservation. Specifically, in order to represent a function with slope 1 almost everywhere, each layer must preserve the norm of the gradient during backpropagation. ReLU architectures satisfy this only when the activations are positive; empirically, this manifests during training of norm-constrained ReLU networks in that the activations are forced to be positive most of the time, reducing the network's capacity to represent nonlinear functions. We make use of an alternative activation function called *GroupSort* — a variant of which was proposed by Chernodub & Nowicki (2016) — which sorts groups of activations. GroupSort is both Lipschitz and gradient norm preserving. Using a variant of the Stone-Weierstrass theorem, we show that norm-constrained GroupSort networks are universal Lipschitz function approximators. While we focus our attention, both theoretically and empirically, on fully connected networks, the same general principles hold for convolutional networks where the techniques we introduce could be directly applied.

Empirically, we show that ReLU networks are unable to solve even the simplest Wasserstein distance estimation problems which GroupSort can solve completely. Moreover, we observe that norm-constrained ReLU networks must trade non-linear processing for gradient norm leading to less expressive networks. We also train classifiers with provable adversarial robustness guarantees and find that using GroupSort provides improved accuracy and robustness compared to ReLU. Across all of our experiments, we found that norm-constrained GroupSort architectures consistently outperformed their ReLU counterparts.

## 2 BACKGROUND

**Notation** We will use $\mathbf{x} \in \mathbb{R}^{in}$ to denote the input vector to the neural network, $\boldsymbol{y} \in \mathbb{R}^{out}$ the output (or logits) of the neural network, $n_l$ the dimensionality of the $l^{th}$ hidden layer, $\mathbf{W}_l \in \mathbb{R}^{n_{l-1} \times n_l}$ and $\boldsymbol{b}_l \in \mathbb{R}^{n_l}$ the weight matrix and the bias of the $l^{th}$ layer. We will denote the pre-activations in layer $l$ with $\boldsymbol{z}_l$ and activations with $\boldsymbol{h}_l$. The number of layers in the network will be $L$ with $\boldsymbol{y} = \boldsymbol{z}_L$. We will use $\phi$ to denote the activation function used in the neural network. The computation performed by layer $l$ of the network will be:

$$\boldsymbol{z}_l = \mathbf{W}_l \boldsymbol{h}_{l-1} + \boldsymbol{b}_l \qquad \boldsymbol{h}_l = \phi(\boldsymbol{z}_l)$$

**Network Jacobian** Using the chain rule, the Jacobian of a neural network can be expanded as follows:

$$\frac{\partial \boldsymbol{y}}{\partial \mathbf{x}} = \frac{\partial \boldsymbol{z}_L}{\partial \boldsymbol{h}_{L-1}} \frac{\partial \boldsymbol{h}_{L-1}}{\partial \boldsymbol{z}_{L-1}} \cdots \frac{\partial \boldsymbol{z}_2}{\partial \boldsymbol{h}^1} \frac{\partial \boldsymbol{h}_1}{\partial \boldsymbol{z}_1} \frac{\partial \boldsymbol{z}_1}{\partial \mathbf{x}} = \mathbf{W}_L \phi'(\boldsymbol{z}_{L-1}) \ldots \mathbf{W}_2 \phi'(\boldsymbol{z}_1) \mathbf{W}_1 \qquad (1)$$

### 2.1 LIPSCHITZ FUNCTIONS

Given two metric spaces $\mathcal{X}$ and $\mathcal{Y}$, a function $f : \mathcal{X} \to \mathcal{Y}$ is Lipschitz continuous if there exists $K \in \mathbb{R}$ such that for all $x_1$ and $x_2$ in $\mathcal{X}$,

$$d_{\mathcal{Y}}(f(x_1), f(x_2)) \leq K d_{\mathcal{X}}(x_1, x_2)$$

where $d_{\mathcal{X}}$ and $d_{\mathcal{Y}}$ are metrics (such as Euclidean distance) on $\mathcal{X}$ and $\mathcal{Y}$ respectively. In this work, when we refer to *the* Lipschitz constant we are referring to the smallest such $K$ for which the above holds under a given $d_{\mathcal{X}}$ and $d_{\mathcal{Y}}$. Unless otherwise specified, we take $\mathcal{X} = \mathbb{R}^n$ and $\mathcal{Y} = \mathbb{R}^m$ throughout. If the Lipschitz constant of a function is $K$, it is called a $K$-*Lipschitz* function. Equivalently, if the function is everywhere differentiable then its Lipschitz constant is bounded by the operator norm of its Jacobian. Throughout this work, we make use of the following definition,

**Definition 1.** *Given a metric space $(X, d_X)$ where $d_X$ denotes the metric on $X$, we write $C_L(X, \mathbb{R})$ to denote the space of all 1-Lipschitz functions mapping $X$ to $\mathbb{R}$ (with respect to the $L_p$ metric).*

### 2.2 LIPSCHITZ-CONSTRAINED NEURAL NETWORKS

As 1-Lipschitz functions are closed under composition, to build a 1-Lipschitz neural network it suffices to compose 1-Lipschitz affine transformations and activation functions.

**1-Lipschitz Linear Transformations:** Ensuring that each linear map is 1-Lipschitz is equivalent to ensuring that $||\mathbf{W}\mathbf{x}||_p \leq ||\mathbf{x}||_p$ for any $\mathbf{x}$; i.e. constraining the matrix $p$-norm, $||\mathbf{W}||_p = \sup_{||\mathbf{x}||_p=1} ||\mathbf{W}\mathbf{x}||_p$, to be at most 1. Important examples of matrix $p$-norms include the matrix 2-norm, which is the largest singular value, and the matrix $\infty$-norm, which can be expressed as:

$$||\mathbf{W}||_\infty = \max_{1 \leq i \leq m} \sum_{j=1}^{m} |w_{ij}|.$$

Similarly, we may also define the mixed matrix norm, given by $||\mathbf{W}||_{p,q} = \sup_{||\mathbf{x}||_p=1} ||\mathbf{W}\mathbf{x}||_q$. Enforcing matrix norm constraints naively may be computationally expensive. Fortunately, techniques exist to efficiently ensure that $||W||_p = 1$ when $p = 2$ or $p = \infty$. We discuss these in Section 4.2.

**1-Lipschitz Activation Functions:** Most commonly used activation functions (such as ReLU (Krizhevsky et al., 2012), sigmoid, tanh, maxout (Goodfellow et al., 2013)) are 1-Lipschitz, if they are scaled appropriately.

### 2.3 APPLICATIONS OF LIPSCHITZ NETWORKS

**Wasserstein Distance Estimation** Wasserstein-1 distance (also called Earth Mover Distance) is a principled distance metric between two probability distributions and has found many applications in machine learning in recent years (Peyré & Cuturi, 2018; Genevay et al., 2017). Using Kantorovich-Rubinstein duality (Villani, 2008), one can recast the Wasserstein distance estimation problem as a concave maximization problem, defined over 1-Lipschitz functions:

$$W(P_1, P_2) = \sup_{f \in C_L(X, \mathbb{R})} \left( \mathbb{E}_{x \sim P_1}[f(x)] - \mathbb{E}_{x \sim P_2}[f(x)] \right) \tag{2}$$

Since this dual objective resembles the discriminator objective for generative adversarial networks (GANs), Arjovsky et al. (2017) proposed the Wasserstein GAN architecture, which uses a neural net architecture to approximate the space of Lipschitz functions.

**Adversarial Robustness**  Adversarial examples are inputs to a machine learning system which have been designed to force undesirable behaviour (Szegedy et al., 2013; Goodfellow et al., 2014). Formally, given a classifier $f$ and a data point $\mathbf{x}$, we write an adversarial example as $\mathbf{x}_{adv} = \mathbf{x} + \delta$ such that $f(\mathbf{x}_{adv}) \neq f(\mathbf{x})$ and $\delta$ is small. A small Lipschitz constant can guarantee a lower bound on the size of $\delta$ (Tsuzuku et al., 2018) and thus provide robustness guarantees. However, existing approaches have both practical and theoretical limitations (Huster et al., 2018).

## 3  GRADIENT NORM PRESERVATION

When backpropagating through a norm-constrained 1-Lipschitz network, the gradient norm is non-increasing as it is processed by each layer. This simple fact leads to interesting consequences when we attempt to represent (scalar-valued) functions whose input-output gradient has norm 1 almost everywhere. (Such functions are relevant to Wasserstein distance estimation, where an optimal dual solution always has this property (Gulrajani et al., 2017).) To ensure the input-output gradient norm is 1, the gradient norm must be preserved by each layer in the network during backpropagation. Unfortunately, norm-constrained networks with common activations are unable to achieve this.

**Theorem 1.** *Consider a neural network, $f : \mathbb{R}^n \to \mathbb{R}$, built with matrix 2-norm constrained weights ($||\mathbf{W}||_2 \leq 1$) and 1-Lipschitz, element-wise, monotonically increasing activation functions. If $||\nabla f(\mathbf{x})||_2 = 1$ almost everywhere, then $f$ is linear.*

As a special case, Theorem 1 shows that no 2-norm-constrained neural network with ReLU (or sigmoid, tanh, etc.) activations can represent the absolute value function. A full proof can be found in Appendix C. Informally, for ReLU layers, the gradient norm can only be preserved if every activation is positive (with the exception of units which don't affect the network's output). But as this holds for almost all inputs, the network's input-output mapping must be linear.

This tension between gradient norm and nonlinear processing is also observed empirically. Figure 5 compares the activation statistics for MNIST classification networks with ReLU activations, with and without matrix norm constraints on the weights. For the network with smallest Lipschitz constant, around 10% of the units are "undead", or always active (and hence do not contribute any nonlinear processing). This suggests that the network is sacrificing nonlinear capacity in order to maintain adequate gradient norm.

Another useful consequence of gradient norm preservation is that we may restrict all of the weight matrices to have singular values of 1:

**Theorem 2.** *Consider a neural network, $f : \mathbb{R}^n \to \mathbb{R}$, built with matrix 2-norm constrained weights and with $||\nabla f(\mathbf{x})||_2 = 1$ almost everywhere. Then, without changing the computed function, each weight matrix $\mathbf{W} \in R^{m \times k}$ can be replaced with a matrix $\widetilde{\mathbf{W}}$ whose singular values all equal 1.*

Note that the condition of singular values equaling 1 is equivalent to the following: when $m > k$, the columns of $\widetilde{\mathbf{W}}$ are orthonormal; when $m < k$, the rows of $\widetilde{\mathbf{W}}$ are orthonormal; and when $m = k$, $\widetilde{\mathbf{W}}$ is orthogonal. For the remainder of this paper, we abuse terminology slightly and refer to such matrices as orthonormal. The proof of Theorem 2 is given in Appendix C. With these two results in place, we restrict our search for expressive Lipschitz networks to those that contain orthonormal weight matrices (those with singular values all equal to 1) and activations which preserve the gradient norm during backpropagation.

## 4  METHODS

We begin by observing that if we can learn any 1-Lipschitz function with a neural network then we can trivially extend this to K-Lipschitz functions by scaling the output by $K$. With this in mind, we focus on designing 1-Lipschitz network architectures with respect to the $L_2$ and $L_\infty$ metrics by requiring *each* layer to be 1-Lipschitz.

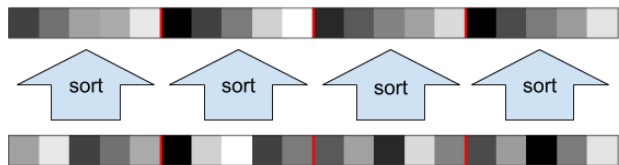

Figure 1: GroupSort activation with a grouping size of 5.

### 4.1 Gradient Norm Preserving Activation Functions

As discussed in Section 3, commonly used activation functions such as ReLU are not gradient norm preserving. To achieve norm preservation, we use a general purpose 1-Lipschitz activation function which we call **GroupSort**. This activation function takes a column vector $\mathbf{x} \in \mathbb{R}^n$, separates the elements into $g$ groups, sorts each group into ascending order, and outputs the combined "group sorted" vector. This is shown graphically in Figure 1.

**Properties of GroupSort**   GroupSort is a Lipschitz operation. Furthermore, it is norm preserving: its Jacobian is a permutation matrix, and permutation matrices preserve every vector $p$-norm. Note also that GroupSort is homogeneous, i.e. $\mathbf{GroupSort}(\alpha\mathbf{x}) = \alpha\mathbf{GroupSort}(\mathbf{x})$, since the sorting order of the elements is invariant to scaling.

**Varying the Grouping Size**   When we pick a grouping size of 2 for GroupSort, we call the operation **MaxMin**. This is equivalent to the Orthogonal Permutation Linear Unit (OPLU) activation (Chernodub & Nowicki, 2016), which was also motivated based on gradient norm preservation. When sorting the *entire* input vector, we call the operation **FullSort**. GroupSort, MaxMin, and Full-Sort are equally expressive, i.e. they can all be reduced to each other, such that the reduction obeys the norm constraint on the weights (for any matrix $p$-norm). We present the details in Appendix A. Compared to MaxMin, FullSort is able to represent certain functions more compactly, but we find that it is typically more difficult to train via stochastic gradient descent.

**Representing other activations**   Under the matrix 2-norm constraint, MaxMin can be seen as equivalent to absolute value. We describe exactly how these activation functions can be transformed into each other in Appendix A. Applying absolute value to the activations has the effect of folding the space on each of the coordinate axes. Hence, a rigid linear transformation, followed by absolute value, followed by another rigid linear transformation, can implement folding along an arbitrary hyperplane. This gives an interesting interpretation of how MaxMin networks can represent certain functions by way of implementing absolute value; an example is shown in Figure 10 in Appendix A. Montufar et al. (2014) provide an in-depth analysis of the expressivity of neural networks built with activations that can perform folding.

Without norm constraints, GroupSort can recover many other common activation functions. For example, ReLU, Leaky ReLU, concatenated ReLU (Shang et al., 2016), and maxout. Details can be found in Appendix A.

### 4.2 Norm-constrained linear maps

We discuss how to practically enforce the 1-Lipschitz constraint on linear layers for 2- and $\infty$-norms.

#### 4.2.1 Enforcing $||W||_2 = 1$ while Preserving Gradient Norm

Several methods have been proposed to enforce matrix 2-norm constraints during training (Cisse et al., 2017; Yoshida & Miyato, 2017). However, in the interest of preserving the gradient norm, we go a step further and enforce *orthonormality* of the weight matrices in each layer. This is a stronger condition, in that we require that all singular values be exactly 1, rather than bounded by 1.

We make use of an algorithm first introduced by Björck & Bowie (1971), which we refer to as Björck Orthonormalization (or simply Björck). Given a matrix, this algorithm finds the closest orthonormal matrix through an iterative application of the Taylor expansion of the polar decomposition. Given an input matrix $A_0 = A$, the algorithm computes,

$$A_{k+1} = A_k \left( I + \frac{1}{2}Q_k + \frac{3}{8}Q_k^2 + \ldots + (-1)^p \binom{-\frac{1}{2}}{p} Q_k^p \right), \tag{3}$$

where $Q_k = I - A_k^T A_k$. Importantly, this algorithm is fully differentiable and thus has a pullback operator for the Stiefel manifold (Absil et al., 2009) allowing us to optimize over orthonormal matrices directly. A larger choice of $p$ adds more computation but gives a closer approximation for each iteration. In practice, we found that we could use $p = 1$ with 2-3 iterations per forward pass and increase this to 15 or more iterations at the end of training to ensure a tightly enforced Lipschitz constraint. We discuss additional details of this algorithm including comparisons to Parseval networks (Cisse et al., 2017) and spectral normalization (Miyato et al., 2018) in Appendix B.

Note that while we focus on fully connected layers, the same general principles apply to convolutions. Convolutions can be *unfolded* and represented as a linear transformation. Up to constant rescaling, the spectral norm of the filter then bounds the spectral norm of the unfolded operation. We do not devote space to computing these constants but instead point readers to other resources which address this question (Gouk et al., 2018; Cisse et al., 2017; Sedghi et al., 2018).

### 4.2.2 ENFORCING $||W||_\infty = 1$

Due to its simplicity and suitability for a GPU implementation, we use Algorithm 1 from Condat (2016) to project the weight matrices onto the $L_\infty$ ball in all of our experiments. Other more sophisticated methods can be found in Condat (2016).

### 4.3 PROVABLE ADVERSARIAL ROBUSTNESS

A small Lipschitz constant limits the change in network output under small adversarial perturbations. As explored by Tsuzuku et al. (2018), we can guarantee adversarial robustness at a point $\mathbf{x}$ by considering the *margin* about that point divided by the Lipschitz constant. Formally, given a network with Lipschitz constant $K$ (with respect to the $L_\infty$ metric) and an input $\mathbf{x}$ with corresponding class $t$ that produces logits $\mathbf{y}$, we define its margin by

$$\mathcal{M}(\mathbf{x}) = \max(0, y_t - \max_{i \neq t} y_i) \tag{4}$$

If $\mathcal{M}(\mathbf{x}) > K\epsilon/2$, then the network is robust to all adversarial perturbations $\delta$ with $||\delta||_\infty < \epsilon$, at $\mathbf{x}$. In this work we train networks with $\infty$-norm constraints on their weights using a multi-class hinge loss:

$$L(\mathbf{y}, t) = \sum_{i \neq t} \max(0, \kappa - (y_t - y_i)) \tag{5}$$

where $\kappa$ controls the margin enforcement and depends on the Lipschitz constant and desired perturbation tolerance (e.g. $\kappa = 0.3 \times K$).

## 5 RELATED WORK

Several methods have been proposed to train Lipschitz neural networks (Cisse et al., 2017; Yoshida & Miyato, 2017; Miyato et al., 2018; Gouk et al., 2018). Cisse et al. (2017) regularize the weights of neural networks to obey an orthonormality constraint and utilize Lipschitz activation functions. In fact, the corresponding update to the weights due to this regularization term can be seen as one step of the Björck orthonormalization scheme (Equation 3). Another approach, spectral normalization (Miyato et al., 2018), employs an efficient implementation of power iteration to rescale each weight by its spectral norm. We compare these methods to Björck orthonormalization in Appendix B. Other researchers (Arjovsky et al., 2016; Wisdom et al., 2016; Sun et al., 2017) have explicitly parameterized square orthogonal weight matrices using, for example, Householder transformations (Householder, 1958).

Other regularization techniques penalize the network Jacobian, thereby constraining the Lipschitz constant locally around the data (Gulrajani et al., 2017; Drucker & Le Cun, 1992; Sokolić et al., 2017). While these methods have the advantage that it is typically easy to train neural networks under such penalties, they do not provably enforce a Lipschitz constraint. Gulrajani et al. (2017) apply the gradient penalty at randomly sampled points between two distributions, but as shown by Gemici et al. (2018), this is often sub-optimal in the context of Wasserstein Distance estimation.

The Lipschitz constant of a neural network has been connected theoretically and empirically to its generalization performance (Bartlett, 1998; Bartlett et al., 2017; Neyshabur et al., 2017; 2018; Sokolić et al., 2017). Neyshabur et al. (2018) show that if the network Lipschitz constant is small then a non-vacuous bound on the generalization error can be derived. Small Lipschitz constants

have also been linked to adversarial robustness (Tsuzuku et al., 2018; Cisse et al., 2017). In fact, adversarial training can be viewed as approximate gradient regularization (Miyato et al., 2017; Simon-Gabriel et al., 2018) which makes the function Lipschitz locally around the training data. Lipschitz constants have been used to provide provable adversarial robustness guarantees. Tsuzuku et al. (2018) manually enforce a margin depending on an approximation of the upper bound on the Lipschitz constant which in turn guarantees adversarial robustness. In this work we also explore provable adversarial robustness through margin training but do so with a network whose Lipschitz constant is known and globally enforced.

Classic neural network universality results use constructions which violate the norm-constraints needed for Lipschitz guarantees (Cybenko, 1989; Hornik, 1991). Huster et al. (2018) explored universal approximation properties of $\infty$-norm-constrained networks and proved that ReLU activations cannot be used to approximate the absolute value function. In this work we also show that many activations, including ReLU, are deficient with 2-norm constraints. However, we prove that Lipschitz functions *can* be universally approximated if the correct activation function is used.

## 6    UNIVERSAL APPROXIMATION OF LIPSCHITZ FUNCTIONS

Universal approximation results for general continuous functions do not directly apply to Lipschitz networks as the constructions typically involve huge Lipschitz constants. Moreover, Huster et al. (2018) showed that it is impossible to approximate even the absolute value function with $\infty$-norm-constrained ReLU networks. In this section, we present theoretical guarantees on the approximation of Lipschitz functions with norm-constrained neural networks. To our knowledge, this is the first universal Lipschitz function approximation result for norm-constrained neural networks.

We will first prove a variant of the Stone-Weierstrass Theorem which gives a simple criterion for universality. (A similar result is presented in Lemma 4.1 in Yaacov (2010).) We then construct a class of networks with the GroupSort activation which satisfy this criterion. We now proceed with the formal statements.

**Definition 2.** *We say that a set of functions, $L$, is a* lattice *if for any $f, g \in L$ we have $max(f, g) \in L$ and $min(f, g) \in L$ (where $max$ and $min$ are defined pointwise).*

**Lemma 1.** *(Restricted Stone-Weierstrass Theorem) Suppose that $(X, d_X)$ is a compact metric space with at least two points and $L$ is a lattice in $C_L(X, \mathbb{R})$ with the property that for any two distinct elements $x, y \in X$ and any two real numbers $a$ and $b$ such that $|a - b| \leq d_X(x, y)$ there exists a function $f \in L$ such that $f(x) = a$ and $f(y) = b$. Then $L$ is dense in $C_L(X, \mathbb{R})$.*

**Remark.** *We could replace $|\cdot|$ with any metric on $\mathbb{R}$.*

The full proof of Lemma 1 is presented in the appendix. Note that Lemma 1 says that $\mathcal{A}$ is a universal approximator for 1-Lipschitz functions if and only if $\mathcal{A}$ is a lattice that separates points. Using Lemma 1, we can derive the second of our key results. Norm-constrained networks with GroupSort activations are able to approximate any Lipschitz function in $L_p$ distance.

**Theorem 3.** *(Universal Approximation with Lipschitz Networks) Let $\mathcal{LN}_p$ denote the class of fully-connected neural networks whose first weight matrix satisfies $||\mathbf{W}_1||_{p,\infty} = 1$, all other weight matrices satisfy $||\mathbf{W}||_\infty = 1$, and MaxMin activations. Let $X$ be a closed and bounded subset of $\mathbb{R}^n$ endowed with the $L_p$ metric. Then the closure of $\mathcal{LN}_p$ is dense in $C_L(X, \mathbb{R})$.*

*Proof.* (Sketch) Observe first that $\mathcal{LN}_p \subset C_L(X, \mathbb{R})$. By Lemma 1, it is sufficient to show that $\mathcal{LN}_p$ is closed under max and min and has the point separation property. For the latter, note that given $x, y \in X$ and $a, b \in \mathbb{R}$ with $|a - b| \leq ||x - y||_p$, we can fit a line with a single layer network, $f$, satisfying the 1-Lipschitz constraint with $f(x) = a$ and $f(x) = b$.

Now consider $f$ and $g$ in $\mathcal{LN}_p$. For simplicity, here assume that they have the same number of layers. We can construct $h \in \mathcal{LN}_\infty$ by taking the weight matrix of the first layer to be the weight matrices of the first layer in $f$ and $g$ vertically concatenated. For the following layers, instead of vertically stacking, we build a block diagonal matrix from the weights of $f$ and $g$. This network is in $\mathcal{LN}_p$ and the final layer of the network outputs $[f(x), g(x)]$. We then apply the GroupSort activation to get $[max(f, g)(x), min(f, g)(x)]$ and finally take the dot product with $[1, 0]$ or $[0, 1]$ to get the max or min respectively. □

We refer readers to Appendix D for the formal proof of Theorem 3 and a diagram of the constructed network in Figure 14. One special case of Theorem 3 is for 1-Lipschitz functions in $L_\infty$ norm,

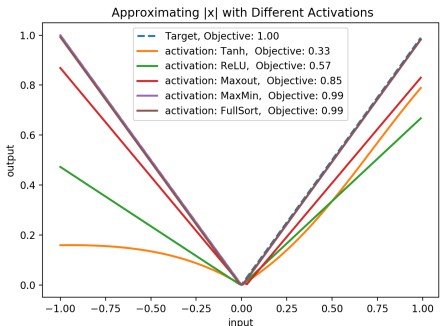

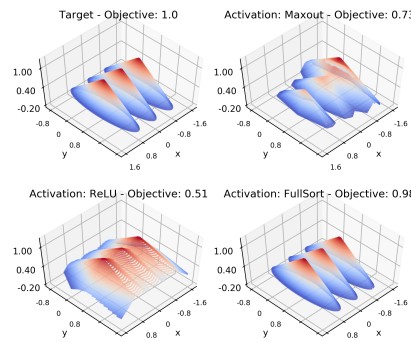

Figure 2: Approximating the absolute value function via Lipschitz networks. The objective values indicate the Wasserstein Distance estimated by each network.

Figure 3: Approximating three circular cones with slope 1 using Lipschitz networks. The objective values indicate the Wasserstein distance estimated by each networks.

where all matrices now satisfy the same constraint: $||W||_\infty = 1$. In this case, we may also extend the restricted Stone-Weierstrass theorem in $L_\infty$ norm to vector-valued functions, and consequently prove universal approximation in this setting. Formally:

**Observation.** *Consider the set of neural networks, $\mathcal{LN}_\infty^m = \{f : \mathbb{R}^n \to \mathbb{R}^m, ||W||_\infty = 1\}$, with MaxMin activations. Then $\mathcal{LN}_\infty^m$ is dense in 1-Lipschitz functions with respect to the $L_\infty$ metric.*

While these constructions rely on the matrix $\infty$-norm of the weight matrices being constrained, we find in practice that constraining the matrix 2-norm makes the networks easier to train, and we have not yet found a Lipschitz function which 2-norm constrained networks have failed to approximate. However, it remains an open question whether 2-norm constrained GroupSort networks are also universal Lipschitz function approximators.

## 7 EXPERIMENTS

Our experiments had two main goals. First, we wanted to test whether the norm-constrained Group-Sort architecture can represent Lipschitz functions other approaches can not. Second, we wanted to test if our networks can perform competitively with existing (heuristic) approaches on practical tasks while maintaining the provable global Lipschitz guarantee. We present additional results in Appendix F, including CIFAR-10 (Krizhevsky, 2009) classification and CelebA (Liu et al., 2015) WGAN training. Other experiment details are found in Appendix G.

### 7.1 REPRESENTATIONAL CAPACITY

In this section, we investigate the ability of 2-norm-constrained networks with different activation functions to represent Lipschitz functions.

#### 7.1.1 QUANTIFYING EXPRESSIVE POWER VIA. WASSERSTEIN DISTANCE ESTIMATION

We propose a simple yet effective method to quantify how expressive different Lipschitz architectures are. We first carefully pick pairs of probability distributions whose Wasserstein Distance and (unique) optimal dual surfaces can be computed analytically. Then, we train neural networks to optimize the dual Wasserstein distance objective (Equation 2) using samples from these distributions and compare the estimated Wasserstein distance and learned dual surfaces to the optimal, analytically computed ones. Expressiveness is measured by how closely the neural network can estimate the correct Wasserstein distance. For 1D and 2D problems, the learned dual surfaces can also be visualized, making it possible to inspect failure modes of non-expressive architectures.

In the following experiments, we trained networks to approximate the absolute value function, multiple two dimensional circular cones and single high dimensional circular cones. Appendix G.1 describes how pairs of probability distributions can be picked which have these optimal dual surfaces, and a Wasserstein distance of precisely 1. In all of the experiments in this section, we use Björck orthonormalization to enforce the 2-norm constraints on the weights.

**Approximating absolute value function:** Figure 2 shows the dual surfaces approximated by Lipschitz-constrained networks with various activation functions. The optimal dual surface is the

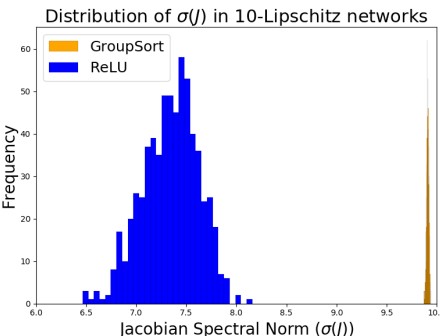
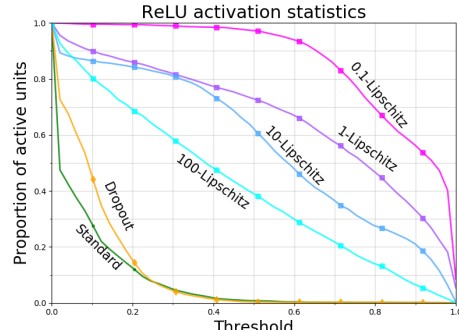

Figure 4: **Jacobian spectral norm distribution** We compare the Jacobian spectral norm of ReLU and GroupSort networks.

Figure 5: **ReLU activation statistics** Ratio of activations which are positive more often than the threshold value on the training data.

absolute value. It can be seen that non-GNP activation functions are incapable of approximating this rather trivial Lipschitz function. While we observed that increasing the network depth helps ReLU and MaxOut activations (Table 1), the representational bottleneck showcased in Figure 2 leads to more severe limitations as the problem dimensionality increases.

**Approximating multiple 2D cones:** Figure 3 shows the dual surfaces approximated by neural networks using various activations. The optimal dual surface is three consecutive circular cones with a gradient of 1 everywhere. Here we observed an even more serious pathology with non-GNP activations: by attempting to increase the slope, the non-GNP networks may distort the shape of the dual surface. When training WGAN critics, this problem cannot be fixed by increasing the Lipschitz constant, since optimal critics for different Lipschitz constants are equivalent up to scaling.

**Approximating high dimensional circular cones:** We evaluated the performance of architectures built with different activation functions for higher dimensional inputs, on the task of approximating high dimensional circular cones which have a gradient of 1 everywhere. As shown in Table 1, this leads to significant drops in the Wasserstein dual objective for Lipschitz networks built with non-GNP activations, and increasing the depth of the networks only slightly improves the situation. We also observed that while the MaxMin activation performs significantly better, it also needs large depth in order to learn the optimal solution. Surprisingly, the FullSort network has no difficulty approximating high dimensional circular cones, even with only two hidden layers.

### 7.1.2 Relevance of Gradient Norm Preservation in Practical Settings

Thus far, we have focused on examples where the gradient of the network should be 1 almost everywhere. But for many practical tasks we do not need to meet this strong condition. Then we should ask, are these pathologies relevant in other settings?

**How much of the Lipschitz capacity can we use?** To understand the practical implications of Theorem 1, we trained two 2-norm-constrained MNIST classifiers scaled to be 10-Lipschitz functions. One with ReLU activations and the other MaxMin. Figure 4 displays the distribution of the largest Jacobian singular value for each network over the training data. Both networks satisfy the Lipschitz constraint but the GroupSort network does so much more tightly than the ReLU network.

| Activ. | Input Dim=128 | | Input Dim=256 | | Input Dim=512 | |
|---|---|---|---|---|---|---|
| Depth | *3* | *7* | *3* | *7* | *3* | *7* |
| ReLU | 0.51 | 0.60 | 0.50 | 0.53 | 0.46 | 0.49 |
| Maxout | 0.66 | 0.71 | 0.60 | 0.66 | 0.52 | 0.56 |
| MaxMin | 0.87 | 0.95 | 0.83 | 0.93 | 0.72 | 0.88 |
| FullSort | **1.00** | **1.00** | **1.00** | **1.00** | **1.00** | **1.00** |

Table 1: **Effect of problem dimensionality on expressiveness:** Testing how well different activation functions and depths can optimize the dual Wasserstein objective with different input dimensionality. The optimal dual surface obtains a dual objective of 1.

| Model | ReLU | Maxout | Maxmin | GroupSort(4) | GroupSort(9) |
|---|---|---|---|---|---|
| **MNIST** | 1.65 | 2.32 | 2.57 | **2.73** | 2.69 |
| **CIFAR-10** | 3.00 | 4.02 | 4.38 | 4.54 | **4.59** |

Table 2: Estimating the Wasserstein Distance between the data and generator distributions using 1-Lipschitz feedforward neural networks, for MNIST and CIFAR-10 GANs.

The ReLU network was not able to make use of the capacity afforded to it and the observed Lipschitz constant was actually closer to 8 than 10. In Appendix F.3 we show the full singular value distribution which suggests that 2-norm-constrained MaxMin networks can achieve dynamical isometry (Pennington et al., 2017) throughout training.

We studied the activation statistics of ReLU networks trained to classify MNIST digits with and without 2-norm constraints in Figure 5. Given a threshold value, $\tau \in [0, 1]$, we computed the proportion of activations throughout the network which are positive at least as often as $\tau$ over the training data distribution. Without a Lipschitz constraint, the activation statistics were very sparse, with almost no units active when $\tau > 0.4$, even when using dropout (Srivastava et al., 2014). When the Lipschitz constraint was enforced the activations were much less sparse with smaller Lipschitz constants amplifying the effect. In the worst case, about 10% of units were "undead", or active all of the time, and hence did not contribute any nonlinear processing. It's not clear what effect this has on the network's representational capacity, but such a dramatic change in the network's activation statistics suggests that it made significant compromises in order to maintain adequate gradient norm.

## 7.2 WASSERSTEIN DISTANCE ESTIMATION

We turn our attention to using norm-constrained GroupSort networks to estimate the Wasserstein distance between the generator distribution of a GAN and the empirical distribution of the data it was trained on. We note that optimal surfaces under the dual Wasserstein objective have a gradient norm of 1 almost everywhere (Corollary 1 in Gemici et al. (2018)). Hence, the gradient norm preservation properties discussed in Section 3 are critical. Appendix G.2 contains details on the experiments described in this section.

### 7.2.1 LOWER BOUNDS ON MNIST AND CIFAR-10 GANS

In this experiment, we first trained a GAN variant on MNIST and CIFAR-10 datasets and then froze the weights of the generator. Using samples from the generator and original data distribution, we trained independent 1-Lipschitz neural networks to compute the Wasserstein distance between the empirical data distribution and the generator distribution. As can be seen in Table 2, using norm-preserving activation functions helps achieve a tighter lower bound on the Wasserstein distance for both MNIST and CIFAR-10 generators.

**Training WGANs** We were also able to train WGANs using our proposed 1-Lipschitz activations and linear transformations. We borrowed the discriminator and generator architectures directly from Chen et al. (2016), but switched the ReLU activations with MaxMin and replaced the standard convolutional and fully connected layers with their Björck counterparts. We also dropped the batch normalization layers, as these would violate the Lipschitz constraint. Figure 6 shows MNIST and CIFAR-10 samples generated using our WGAN variant. We leave further investigation of the WGANs built with our techniques to a future study.

### 7.3 ROBUSTNESS AND INTERPRETABILITY OF LIPSCHITZ NETWORKS

We explored the robustness of Lipschitz neural networks trained on MNIST to adversarial perturbations measured with $L_\infty$ distance. When training the networks we enforced an $L_\infty$ constraint on the weights and used the multi-class hinge loss from Equation 5. We found this to be more effective than the manual margin training used by Tsuzuku et al. (2018). We trained all networks with a Lipschitz constant of $K = 1000$ and chose the margin $\kappa = Ka$ where $a$ was 0.1 or 0.3. Notably, this technique provides margin-based provable robustness guarantees as described in Section 4.3. We then attacked these models using the FGS and PGD methods (Szegedy et al., 2013; Madry et al., 2017) under the CW loss (Carlini & Wagner, 2016). The results are presented in Table 3 and Figure 8. The Lipschitz networks with MaxMin activations were able to achieve better clean accuracy and larger margins than their ReLU counterparts which led to considerably improved adversarial robustness.

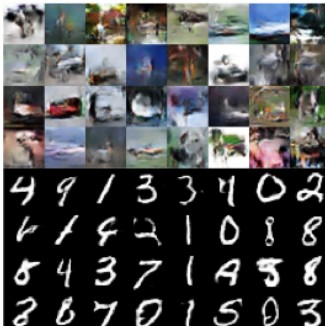

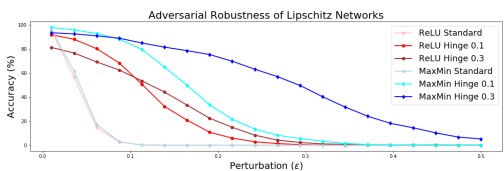

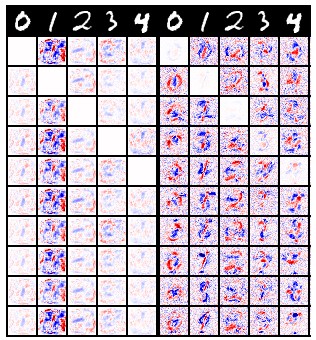

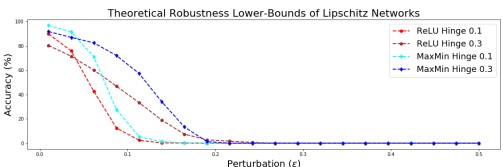

Figure 6: Samples from WGANs whose critic architectures were built using GNP atomic units.

Figure 7: Gradients of input images with respect to targeted cross-entropy loss. Left: standard network, Right: 2-norm-constrained network.

Figure 8: **Adversarial Robustness** Accuracy on PGD adversarial examples for varying perturbation sizes $\epsilon$.

Figure 9: **Theoretical Adversarial Robustness** Theoretical accuracy lower bound for varying perturbation sizes $\epsilon$.

With the strictly enforced Lipschitz constant, we can compute theoretical lower bounds on the accuracy against adversaries with a maximum perturbation strength $\epsilon$. In Figure 9, we show this lower bound for each of the models previously studied. This is computed by finding the proportion of data points which violate the margin by at least $K\epsilon$. Note that at the computed threshold, the model has low confidence in the adversarial example. An even larger perturbation would be required to induce confident misclassification.

Tsipras et al. (2018) reported that networks trained using adversarial training learn robust features which allow them to have interpretable gradients. We found that the same is true for Lipschitz networks, even without using adversarial training. The gradients with respect to the inputs are displayed for a standard network and a 2-norm-constrained network in Figure 7. The first row shows the original images with following rows showing the gradient with different class targets (0-9). Positive pixel values are red and blue is negative.

| Model | Clean | FGS | | PGD | |
|---|---|---|---|---|---|
| | Err. | $\epsilon = 0.1$ | $\epsilon = 0.3$ | $\epsilon = 0.1$ | $\epsilon = 0.3$ |
| Standard ReLU | 1.61 | 78.91 | 98.54 | 99.81 | 100.0 |
| Standard MaxMin | **1.47** | 79.60 | 99.81 | 99.91 | 100.0 |
| Margin-0.1 ReLU | 5.48 | 48.54 | 99.52 | 76.07 | 100.0 |
| Margin-0.1 MaxMin | 1.92 | 22.85 | 99.61 | 40.23 | 98.93 |
| Margin-0.3 ReLU | 15.20 | 46.49 | 98.28 | 61.33 | 100.0 |
| Margin-0.3 MaxMin | 5.02 | **14.16** | **51.57** | **15.14** | **59.67** |

Table 3: **Adversarial robustness** Classification error for varying $L_\infty$ distance of adversarial attacks.

## 8 CONCLUSION

We have identified gradient norm preservation as a critical component of Lipschitz network design and showed that failure to achieve this leads to less expressive networks. By combining the Group-Sort activation function and orthonormal weight matrices, we presented a class of neural networks which are provably 1-Lipschitz and can approximate any 1-Lipschitz function arbitrarily well. Empirically, we showed that our GroupSort networks are more expressive than existing architectures and can be used to achieve better estimates of Wasserstein distance and provable adversarial robustness guarantees.

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

# Appendices

## A    GROUPSORT ACTIVATION

**FullSort and MaxMin**    FullSort can implement MaxMin by simply "chunking" the biases in pairs. To be more precise, let $x_{max} = \sup_{\mathbf{x} \in \mathcal{X}} ||\mathbf{x}||_\infty$ where $\mathcal{X}$ represents the domain, and $\boldsymbol{b} = [b_1, b_2, ..., b_n]^T$ where $x_{max} < b_1 = b_2 \ll b_3 = b_4 \ll \cdots \ll b_{n-1} = b_n$ ($\ll$ denotes differing by at least $x_{max}$). We can write:

$$\mathbf{MaxMin(x)} = \mathbf{FullSort(Ix + \boldsymbol{b})} - \boldsymbol{b},$$

where $\mathbf{I}$ denotes the identity matrix. Similarly, FullSort can be represented using a series of MaxMin layers that implement BubbleSort; note that this construction obeys any matrix $p$-norm constraint since it can be implemented using only permutation matrices for the weights.

**MaxMin and absolute value**    MaxMin and absolute value can each represent eachother under 2-norm-constrained weights. The two operations are reduced to each other as follows:

$$\begin{bmatrix} \mathbf{max}(x) \\ \mathbf{min}(y) \end{bmatrix} = \begin{bmatrix} \frac{1}{\sqrt{2}} & \frac{1}{\sqrt{2}} \\ \frac{1}{\sqrt{2}} & \frac{-1}{\sqrt{2}} \end{bmatrix} \mathbf{abs}(\begin{bmatrix} \frac{1}{\sqrt{2}} & \frac{1}{\sqrt{2}} \\ \frac{1}{\sqrt{2}} & \frac{-1}{\sqrt{2}} \end{bmatrix} \begin{bmatrix} x \\ y \end{bmatrix} + \begin{bmatrix} B \\ 0 \end{bmatrix}) - \begin{bmatrix} \sqrt{2}B \\ 0 \end{bmatrix} \quad (6)$$

$$\mathbf{abs}(x) = \begin{bmatrix} \frac{1}{\sqrt{2}} & -\frac{1}{\sqrt{2}} \end{bmatrix} \mathbf{MaxMin}(\begin{bmatrix} \frac{1}{\sqrt{2}} \\ \frac{-1}{\sqrt{2}} \end{bmatrix} x) \quad (7)$$

In Equation 6, the value of $B$ is chosen such that $2\mathbf{x} + \sqrt{2}B > 0$ for all $\mathbf{x}$ in the domain. Note that all the matrices in these constructions satisfy the matrix 2-norm constraint.

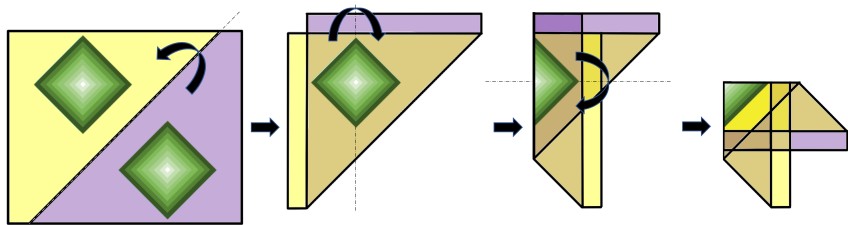

Figure 10: A rigid linear transformation, followed by absolute value, followed by another rigid linear transformation, can implement folding along an arbitrary hyperplane. Here is an example where the network represents a function consisting of a pair of square pyramids by folding the space three times, until the function is representable as a linear function of the top layer activations.

**GroupSort and other activations**    Here we show that GroupSort can recover ReLU, maxout, and concatenated ReLU activation functions. We first show that MaxMin can recover ReLU and concatenated ReLU. Note that,

$$\mathbf{MaxMin}(\begin{bmatrix} x \\ 0 \end{bmatrix}) = \begin{bmatrix} ReLU(x) \\ -ReLU(-x) \end{bmatrix} \quad (8)$$

Thus, by adding $0$ elements to the pre-activations and then applying another linear transformation after MaxMin we can output either ReLU or concatenated ReLU. If instead of adding $0$ to the preactivations we added $ax$ we could recover Leaky ReLU by using a linear transformation to select $\max(x, ax)$.

To recover maxout with groups of size $k$, we perform GroupSort with groups of size $k$ and use the next linear transformation to select the first element of each group after sorting (corresponding to the max).

## B    IMPLEMENTING NORM CONSTRAINTS

When implementing the norm constraints it is possible to project the weight matrices *after* each gradient descent step, or *during* the forward pass (if the projection is differentiable). For the Björck algorithm we utilize the latter while Parseval networks use the Björck algorithm after each gradient descent step. In any case, once training has completed we can project the weights to enforce the norm constraint and use these as our fixed weights at test time - removing the computational overhead required during training.

### B.1    COMPARING BJÖRCK AND PARSEVAL

In Cisse et al. (2017), the authors motivate an update to the weight matrices by considering the gradient of a regularization term, $\frac{\beta}{2}||W^TW - I||_F^2$. By subtracting this gradient from the weight matrices they push them closer to the Stiefel manifold. The final update is given by,

$$W \leftarrow W(I + \beta) - \beta WW^TW \tag{9}$$

Note that when $\beta = 0.5$ this update is exactly the first order ($p = 1$) update from Equation 3, with a single iteration. Compared to our approach, the key difference in Parseval networks is that the weight matrix update is applied *after* the primary gradient update. For our approach, we utilize the algorithm in Equation 3 during the network forward pass to optimize directly on the Stiefel manifold. This is more expensive but lets us ensure that the weight matrices are close to orthonormal throughout training.

**Choice of** $\beta$    We can relate the first order Björck algorithm to the Parseval update by setting $\beta = 0.5$. However, in practice Parseval networks are trained with very small choices of $\beta$, for example $\beta = 0.0003$. As expected, when $\beta$ is small the algorithm still converges to an orthonormal matrix but much more slowly. Figure 11 shows the maximum and minimum singular values of matrices which have undergone 50 iterations of the first order Björck scheme for varying choices of $\beta < 0.5$. When $\beta$ is much smaller than $0.5$ the matrices may be far from orthonormal. We also show how the maximum and minimum singular values vary over the number of iterations when $\beta = 0.0003$ (a common choice for Parseval networks) in Figure 12. This has practical implications for Parseval training, particularly when using early stopping, as the weight matrices may be far from orthonormal if the gradients are relatively large compared to the update produced by the Björck algorithm. We observed this effect empirically in our MNIST classification experiments but found that Parseval networks were still able to achieve a meaningful regularization effect.

### B.2    COMPARING BJÖRCK AND SPECTRAL NORMALIZATION

Spectral Normalization (Miyato et al., 2018) enforces the largest singular value of each weight matrix to be less than 1 by estimating the largest singular value and left/right singular vectors using power iteration, and normalizing the weight matrix using these during each forward pass. While this constraint does allow *all* singular values of the weight matrix to be 1, we have found that this rarely happens in practice. Hence, enforcing the 1-Lipschitz constraint via spectral normalization doesn't guarantee gradient norm preservation.

We demonstrate the practical consequences of the inability of spectral normalization to preserve gradient norm on the task of approximating high dimensional cones. In order to quantify approximation performance, we carefully pick two $n$ dimensional probability distributions such that 1) The Wasserstein Distance between them is exactly 1 and 2) the optimal dual surface consists of an $n - 1$ dimensional cones with a gradient of 1 everywhere, embedded in $n$ dimensions. We trained 1-Lipschitz constrained neural networks to optimize the dual Wasserstein objective in 2 and checked how well the architecture of choice is able to approximate the optimal dual surface, measured by the Wasserstein Distance they estimate. Please refer to Section 7.1.1 for more experiments in this flavor and Appendix G.1 for how these two probability distributions are picked.

Figure 13 shows that neural networks trained with Björck orthonormalization not only are able to approximate high dimensional cones better than spectral normalization, but also converge much faster in terms of training iterations. The gap between these methods gets much more significant as the problem dimensionality increases. In this experiment, each network consisted of 3 hidden layers

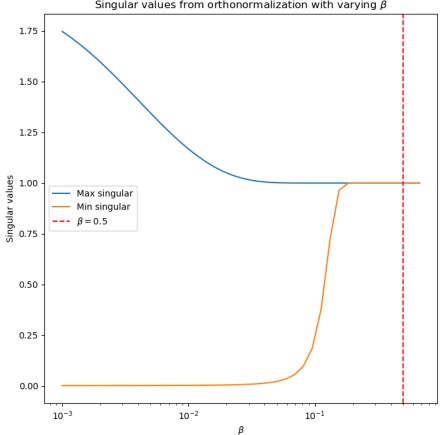 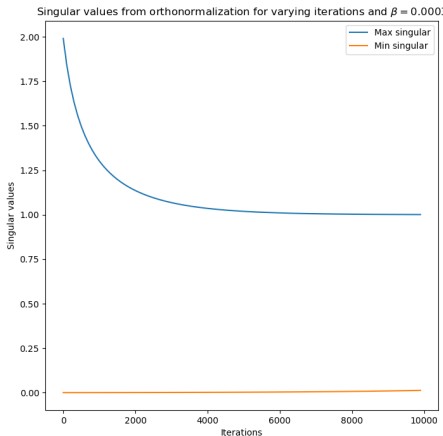

Figure 11: Convergence of the Björck algorithm for different choices of $\beta$. The largest and smallest singular values are shown after 50 iterations of the algorithm.

Figure 12: Convergence of the Björck algorithm for increasing iterations with $\beta = 0.0003$. The largest and smallest singular values are shown after each iteration of the algorithm.

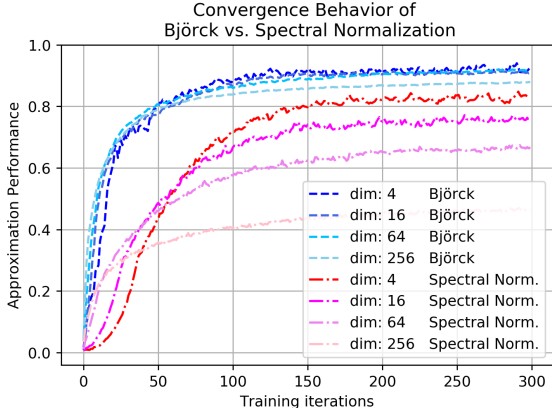

Figure 13: Comparing the performance of 1-Lipschitz neural nets using Björck orthonormalization and spectral normalization to enforce the 2-norm constraint on the high dimensional cone fitting task (Section 7.1.1). Note that networks using Björck orthonormalization both converge faster and achieve higher final approximation accuracies, as measured by the estimated Wasserstein Distance.

with 512 hidden units per layer, and was trained with the Adam optimizer (Kingma & Ba, 2014) with its default hyperparameters. Tuned learning rates of 0.01 for Björck and 0.0033 for spectral normalization were used.

### B.3    SUFFICIENT CONDITION FOR CONVERGENCE OF BJÖRCK ORTHONORMALIZATION

The Björck orthonormalization can be shown to always converge as long as the condition $||\mathbf{W}^T\mathbf{W} - \mathbf{I}||_2 < 1$ is satisfied (Hasenclever et al.). When viewed in conjunction with the fact that the output of this procedure is scale-invariant ($\mathbf{BJORCK}(\alpha\mathbf{W}) = \alpha\mathbf{BJORCK}(\mathbf{W})$) (Björck & Bowie, 1971), the aforementioned sufficient condition can be implemented by simply scaling the weight matrix so that all of its singular values are smaller than or equal to 1 before orthonormalization.

A scaling factor can be computed efficiently by considering the following matrix norm inequalities:

$$\sigma_{max} \leq \sqrt{m * n} ||\mathbf{W}||_{max} \tag{10}$$

$$\sigma_{max} \leq \sqrt{n} ||\mathbf{W}||_1 \tag{11}$$

$$\sigma_{max} \leq \sqrt{m} ||\mathbf{W}||_\infty \tag{12}$$

Above, $\sigma_{max}$ corresponds to the largest singular value of the matrix and $m$ and $n$ stand for the number of rows and columns respectively. Note that computing the quantities on the right hand side of the inequalities involves at most summing over the rows or columns of the weight matrix, which is a cheap operation.

## C  NON-EXPRESSIVE NORM-CONSTRAINED NETWORKS ARE LINEAR

**Theorem 1.** *Consider a neural network, $f : \mathbb{R}^n \to \mathbb{R}$, built with matrix 2-norm constrained weights ($||\mathbf{W}||_2 \leq 1$) and 1-Lipschitz, element-wise, monotonically increasing activation functions. If $||\nabla f(\mathbf{x})||_2 = 1$ almost everywhere, then $f$ is linear.*

*Proof.* We can express the input-output Jacobian of a neural network as:

$$\frac{\partial f}{\partial \mathbf{x}} = \frac{\partial f}{\partial \boldsymbol{h}_{L-1}} \frac{\partial \boldsymbol{h}_{L-1}}{\partial \boldsymbol{z}_{L-1}} \frac{\partial \boldsymbol{z}_{L-1}}{\partial \mathbf{x}} = \mathbf{W}_L \frac{\partial \phi(\boldsymbol{z}_{L-1})}{\partial \boldsymbol{z}_{L-1}} \frac{\partial \boldsymbol{z}_{L-1}}{\partial \mathbf{x}}$$

Note that $\mathbf{W}_L \in \mathbb{R}^{1 \times n_{L-1}}$. Moreover, using the sub-multiplicativity of matrix norms, we can write:

$$1 = \left\|\frac{\partial f}{\partial \mathbf{x}}\right\|_2 \leq \left\|\mathbf{W}_L \frac{\partial \phi(\boldsymbol{z}_{L-1})}{\partial \boldsymbol{z}_{L-1}}\right\|_2 \left\|\frac{\partial \boldsymbol{z}_{L-1}}{\partial \mathbf{x}}\right\|_2 \leq ||\mathbf{W}_L||_2 \left\|\frac{\partial \phi(\boldsymbol{z}_{L-1})}{\partial \boldsymbol{z}_{L-1}}\right\|_2 \left\|\frac{\partial \boldsymbol{z}_{L-1}}{\partial \mathbf{x}}\right\|_2 \leq 1$$

for $x$ almost everywhere. The quantity is also upper bounded by 1 due to the 1-Lipschitz property. Therefore, all of the Jacobian norms in the above equation must be equal to 1. Notably,

$$\left\|\mathbf{W}_L \frac{\partial \phi(\boldsymbol{z}_{L-1})}{\partial \boldsymbol{z}_{L-1}}\right\|_2 = 1 \quad \text{and} \quad ||\mathbf{W}_L||_2 = 1$$

We then consider the following operation:

$$||\mathbf{W}_L||_2^2 - \left\|\mathbf{W}_L \frac{\partial \phi(\boldsymbol{z}_{L-1})}{\partial \boldsymbol{z}_{L-1}}\right\|_2^2 = \sum_{i=1}^n (1 - \left(\frac{\partial \phi(\boldsymbol{z}_{L-1})}{\partial \boldsymbol{z}_{L-1}}\right)_{ii}^2)(W_{L,i})^2 = 0 \tag{13}$$

We have $0 \leq \frac{\partial \phi}{\partial \boldsymbol{z}_L} \leq 1$ as $\phi$ is 1-Lipschitz and monotonically increasing. Therefore, we must have either $\frac{\partial \phi}{\partial \boldsymbol{z}_L}{}_{ii} = 1$ almost everywhere, or $\mathbf{W}_{L,i} = 0$. Thus we can write,

$$\boldsymbol{z}_L = \sum_{i=1}^m W_{L,i} \phi(\boldsymbol{z}_{L-1})_i + b_L = \sum_{i:W_{L,i} \neq 0} W_{L,i} \phi(\boldsymbol{z}_{L-1})_i + b_L$$

$$= \sum_{i:W_{L,i} \neq 0} W_{L,i} \boldsymbol{z}_{L-1,i} + b_L$$

Then $\boldsymbol{z}_L$ can be written as a linear function of $\boldsymbol{z}_{L-1}$ almost everywhere and by Lipschitz continuity we must in fact have that $\boldsymbol{z}_L$ *is* a linear function of $\boldsymbol{z}_{L-1}$. In particular, we can write $\boldsymbol{z}_L = \mathbf{W}_L \mathbf{W}_{L-1} \boldsymbol{h}_{L-2} + (\mathbf{W}_L \boldsymbol{b}_{L-1} + b_L)$, thus collapsing the last two layers into a single linear layer, with weight matrix $\mathbf{W}_L \mathbf{W}_{L-1} \in \mathbb{R}^{1 \times n_{L-2}}$ and scalar bias $\mathbf{W}_L \boldsymbol{b}_{L-1} + b_L$.

From here we can apply the exact same argument as above to $\phi(\mathbf{z}_{L-2})$, reducing the next layer to be linear. By repeating this all the way to the first linear layer we collapse the network into a single linear function. □

**Theorem 2.** *Consider a neural network, $f : \mathbb{R}^n \to \mathbb{R}$, built with matrix 2-norm constrained weights and with $||\nabla f(\mathbf{x})||_2 = 1$ almost everywhere. Then, without changing the computed function, each weight matrix $\mathbf{W} \in R^{m \times k}$ can be replaced with a matrix $\widetilde{\mathbf{W}}$ whose singular values all equal 1.*

*Proof.* Take a weight matrix $\mathbf{W}_i$, for $i < L$. By the argument presented in the proof of Theorem 1, this weight matrix must preserve the norm of gradients during backpropagation. That is,

$$1 = \left\| \frac{\partial f}{\partial \mathbf{z}_i} \mathbf{W}_i \right\|_2$$

Using the singular value decomposition, we write $\mathbf{W}_i = \mathbf{U}\mathbf{\Sigma}\mathbf{V}^T$. We then define $\widetilde{\mathbf{W}}_i = \mathbf{U}\widetilde{\mathbf{\Sigma}}\mathbf{V}^T$ where $\tilde{\mathbf{\Sigma}}$ has ones along the diagonal. Furthermore, define $\mathbf{W}_i^{(t)} = t\mathbf{W}_i + (1-t)\widetilde{\mathbf{W}}_i$. Now replace $\mathbf{W}_i$ with $\mathbf{W}_i^{(t)}$ in the network. Then we have,

$$\frac{\partial f}{\partial t} = \frac{\partial f}{\partial \mathbf{z}_i} \frac{\partial \mathbf{z}_i}{\partial t} = \frac{\partial f}{\partial \mathbf{z}_i}(\mathbf{W}_i - \widetilde{\mathbf{W}}_i)\boldsymbol{h}_{i-1} = \frac{\partial f}{\partial \mathbf{z}_i}\mathbf{U}(\mathbf{\Sigma}_i - \widetilde{\mathbf{\Sigma}}_i)\mathbf{V}^T\boldsymbol{h}_{i-1}$$

As the norm of $\frac{\partial f}{\partial \mathbf{z}_i}$ is preserved by $\mathbf{W}_i$ we must have that $\boldsymbol{u} = (\frac{\partial f}{\partial \mathbf{z}_i}\mathbf{U})^T$ has non-zero entries only where the diagonal of $\mathbf{\Sigma}$ is 1. That is, $u_j = 0 \iff \mathbf{\Sigma}_{jj} < 1$. In particular, we have $\boldsymbol{u}^T\mathbf{\Sigma}_i = \boldsymbol{u}^T\widetilde{\mathbf{\Sigma}}_i$ meaning $\frac{\partial f}{\partial t} = 0$. Thus, the output of the network is the same for all $t$, in particular for $t = 0$ and $t = 1$. Thus, we can replace $\mathbf{W}_i$ with $\widetilde{\mathbf{W}}_i$ and the network output remains unchanged.

We can repeat this argument for all $i < L$ (for $i = 1$ we adopt the notation $\boldsymbol{h}_0 = \boldsymbol{x}$, the input to the network). For $i = L$ the result follows directly. $\qquad\square$

## D    UNIVERSAL APPROXIMATION OF 1-LIPSCHITZ FUNCTIONS

Here we present formal proofs related to finding neural network architectures which are able to approximate any 1-Lipschitz function. We begin with a proof of Lemma 1.

**Lemma 1.** *(Restricted Stone-Weierstrass Theorem) Suppose that $(X, d_X)$ is a compact metric space with at least two points and $L$ is a lattice in $C_L(X, \mathbb{R})$ with the property that for any two distinct elements $x, y \in X$ and any two real numbers $a$ and $b$ such that $|a - b| \leq d_X(x, y)$ there exists a function $f \in L$ such that $f(x) = a$ and $f(y) = b$. Then $L$ is dense in $C_L(X, \mathbb{R})$.*

*Proof.* This proof follows a standard approach with small modifications. We aim to show that for any $g \in C_L(X, \mathbb{R})$ and $\epsilon > 0$ we can find $f \in L$ such that $||g - f||_\infty < \epsilon$ (i.e. the largest difference is $\epsilon$).

Fix $x \in X$. Then for each $y \in X$, we have an $f_y \in L$ with $f_y(x) = g(x)$ and $f_y(y) = g(y)$. This follows from the separation property of $L$ and, using the fact that $g$ is 1-Lipschitz, $|g(x) - g(y)| \leq d_X(x, y)$.

Define $V_y = \{z \in X : f_y(z) < g(z) + \epsilon\}$. Then $V_y$ is open and we have $x, y \in V_y$. Therefore, the collection of sets $\{V_y\}_{y \in X}$ is an open cover of $X$. By the compactness of $X$, there exists some finite subcover of $X$, say, $\{V_{y_1}, \ldots, V_{y_n}\}$, with corresponding functions $f_{y_1}, \ldots, f_{y_n}$.

Let $F_x = min(f_{y_1}, \ldots, f_{y_n})$. Since $L$ is a lattice we must have $F_x \in L$. And moreover, we have that $F_x(x) = g(x)$ and $F_x(z) < g(z) + \epsilon$, for all $z \in X$.

Now, define $U_x = \{z \in X : F_x(z) > g(z) - \epsilon\}$. Then $U_x$ is an open set containing $x$. Therefore, the collection $\{U_x\}_{x \in X}$ is an open cover of $X$ and admits a finite subcover, $\{U_{x_1}, \ldots, U_{x_m}\}$, with corresponding functions $F_{x_1}, \ldots, F_{x_m}$.

Let $G = max(F_{x_1}, \ldots, F_{x_m}) \in L$. We have $G(z) > g(z) - \epsilon$, for all $z \in X$.

Combining both inequalities, we have that $g(z) - \epsilon < G(z) < g(z) + \epsilon$, for all $z \in X$. Or more succinctly, $||g - G||_\infty < \epsilon$. The result is proved by taking $f = G$. $\qquad\square$

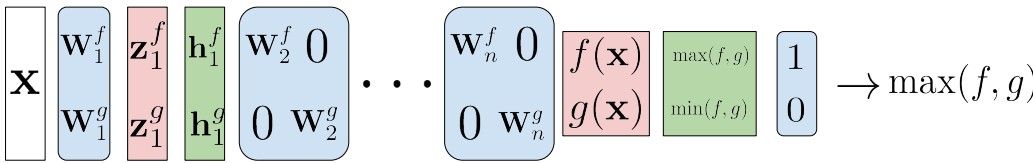

Figure 14: Lattice construction for $L_p$ universal approximation.

We now proceed to prove Theorem 3.

**Theorem 3.** *(Universal Approximation with Lipschitz Networks) Let $\mathcal{LN}_p$ denote the class of fully-connected neural networks whose first weight matrix satisfies $||\mathbf{W}_1||_{p,\infty} = 1$, all other weight matrices satisfy $||\mathbf{W}||_\infty = 1$, and MaxMin activations. Let $X$ be a closed and bounded subset of $\mathbb{R}^n$ endowed with the $L_p$ metric. Then the closure of $\mathcal{LN}_p$ is dense in $C_L(X, \mathbb{R})$.*

*Proof.* The first property we require is separation of points. This follows trivially as given four points satisfying the required conditions we can find a linear map with the required $L_{p,\infty}$ matrix norm that fits them. It remains then to prove that we can construct a lattice under this constraint. We begin by considering two 1-Lipschitz neural networks, $f$ and $g$. We wish to design an architecture which is guaranteed to be 1-Lipschitz and can represent both $\max(f,g)$ and $\min(f,g)$.

The key insight we will use is the idea that we can split the network into two parallel *channels* which each computes one of $f$ and $g$. At the end of the network, we can then select one of these channels depending on whether we want the max or the min.

Each of the networks $f$ and $g$ is determined by a set of weights and biases, we will denote these $[\mathbf{W}_1^f, \mathbf{b}_1^f, \ldots, \mathbf{W}_n^f, b_n^f]$ and $[\mathbf{W}_1^g, \mathbf{b}_1^g, \ldots, \mathbf{W}_n^g, \mathbf{b}_n^g]$ for $f$ and $g$ respectively. For now, assume that these networks are of equal depth (we can lift this assumption later) however we make no assumptions on the width. We will now construct $h = max(f,g)$ in the form of a 1-Lipschitz neural network. To achieve this, we will design a network $h$ which first concatenates the first layers of networks $f$ and $g$ and then computes $f$ and $g$ separately before combining them at the end.

We take the first weight matrix of $h$ to be $\mathbf{W}_1^h = [\mathbf{W}_1^f \ \mathbf{W}_1^g]^T$, that is the weight matrices of $f$ and $g$ stacked vertically. This matrix necessarily satisfies $||\mathbf{W}_1^h||_{p,\infty} = 1$. Similarly, the bias will be those from the first layers of $f$ and $g$ stacked vertically. Then the first layer's pre-activations will be exactly the pre-activations of $f$ and $g$ stacked vertically.

For the following layers, we construct the biases in the same manner (vertical stacking). We construct the weights by constructing new block-diagonal weight matrices. That is, given $\mathbf{W}_i^f$ and $\mathbf{W}_i^g$, we take

$$W_i^h = \left[ \begin{array}{cc} W_i^f & 0 \\ 0 & W_i^g \end{array} \right]$$

This matrix also has $\infty$-norm equal to 1. We repeat this for each of the layers in $f$ and $g$ and end up with a final layer which has two units, $f$ and $g$. We can then take MaxMin of this final layer and take the inner product with $[1, 0]$ to recover the max or $[0, 1]$ for the min.

Finally, we must address the case where the depth of $f$ and $g$ are different. In this case we notice that we are able to represent the identity function with MaxMin activations. To do so observe that after the pre-activations have been sorted we can multiply by the identity and the sorting activation afterwards will have no additional effect. Therefore, for the channel that has the smallest depth we can add in these additional identity layers to match the depths and resort to the above case.

We have shown that the set of neural networks is a lattice which separates points, and thus by Lemma 1 it must be dense in $C_L(X, \mathbb{R})$. □

Note that we could have also used the maxout activation Goodfellow et al. (2013) to complete this proof. This makes sense, as the maxout activation is also norm-preserving in $L_\infty$. However, this

does not hold when using a 2-norm constraint on the weights. We now present several consequences of the theoretical results given above.

This result can be extended easily to vector-valued Lipschitz functions with respect to $L_\infty$ distance by noticing that the space of such 1-Lipschitz functions is a lattice. We may apply the Stone-Weierstrass proof to each of the coordinate functions independently and use the same construction as in Theorem 3 modifying only the last layer which will now reorder the outputs of each function to do a pairwise comparison and then select the relevant components to produce the max or the min.

**Observation.** *Consider the set of neural networks, $\mathcal{LN}_\infty^m = \{f : \mathbb{R}^n \to \mathbb{R}^m, ||W||_\infty = 1\}$, with MaxMin activations. Then $\mathcal{LN}_\infty^m$ is dense in 1-Lipschitz functions with respect to the $L_\infty$ metric.*

*Proof.* Note that given two functions, $g, f : \mathbb{R}^n \to \mathbb{R}^m$ which are 1-Lipschitz with respect to the $L_\infty$ metric, their element-wise max (or min) is also 1-Lipschitz with respect to the $L_\infty$ metric. Consider the element-wise components of such an $f$, written $f = (f_1, \ldots, f_m)$. We can apply the Stone-Weierstrass theorem (Lemma 1) to each of the components independently, such that if the same conditions apply (trivially extended to $\mathbb{R}^m$) the Lattice is dense. Thus, as in the proof of Theorem 3, it suffices to find a network $h \in \mathcal{LN}_\infty^m$ which can represent the max or min of any other networks, $f, g \in \mathcal{LN}_\infty^m$.

In fact, we can use almost exactly the same construction as in the proof of Theorem 3. We follow the same initial steps by concatenating weight matrices and constructing block-diagonal matrices from the two networks. After doing this for all layers in the networks $f$ and $g$, we will output $[f_1, \ldots, f_m, g_1, \ldots g_m]$. We can then permute these entries using a single linear layer to produce $[f_1, g_1, f_2, g_2, \ldots, f_m, g_m]$ finally we take MaxMin and use the final weight matrix to select either $\max(f, g)$ or $\min(f, g)$. □

# E    SPECTRAL JACOBIAN REGULARIZATION

Most existing work begins with the goal of constraining the spectral norm of the Jacobian and proceeds to achieve this by placing constraints on the weights of the network (Yoshida & Miyato, 2017). While not the main focus of our work, we propose a simple new technique which allows us to directly regularize the spectral norm of the Jacobian, $\sigma(J)$. This method differs from the ones described previously as the Lipschitz constant of the entire network is regularized using a single term, instead of at the layer level.

The intuition for this algorithm follows that of Yoshida & Miyato (2017), who apply power iteration to estimate the singular values of the weight matrices online. The authors also discuss computing the spectral radius of the Jacobian directly, and related quantities such as the Frobenius norm, but dismiss this as being too computationally expensive.

Power iteration can be used to compute the leading singular value of a matrix $J$ with the following repeated steps,

$$\mathbf{v}_k = J^T \mathbf{u}_{k-1} / ||J^T \mathbf{u}_{k-1}||_2, \mathbf{u}_k = J\mathbf{v}_k / ||J\mathbf{v}_k||_2$$

Then we have $\sigma(J) \approx \mathbf{u}^T J \mathbf{v}$. There are two challenges that must be overcome to implement this in practice. First, the algorithm requires higher order derivatives which leads to increased computational overhead. However, the tradeoff is often reasonable in practice, see e.g. Drucker & Le Cun (1992). Second, the algorithm requires both Vector-Jacobian products and Jacobian-Vector products. The former can be computed with reverse-mode automatic differentiation but the latter requires the less common forward-mode. Fortunately, one can recover forward-mode from reverse mode by constructing Vector-Jacobian products and utilizing the transpose operator (Townsend, 2017). In this setting, we can actually re-use the intermediate reverse-mode backpropagation within the algorithm which further reduces the computational overhead. The algorithm itself is presented as Algorithm 1.

We present this algorithm primarily to be used for regularization but this could also be used to approximately control the Lipschitz constraint by rescaling the output of the entire network by the estimate of the Jacobian spectral norm in a similar fashion to weight spectral normalization Miyato et al. (2018).

---

**Algorithm 1:** Spectral Jacobian Regularization

Initialize $u$ randomly, choose hyperparameter $\lambda > 0$

**for** *data batch (X,Y)* **do**

    Compute logits $f_\theta(X)$

    Compute loss $\mathcal{L}(f_\theta(X), Y)$

    Compute $\mathbf{g} = \mathbf{u}^T \dfrac{\partial \mathbf{f}}{\partial \mathbf{x}}$, using reverse mode

    Set $\mathbf{v} = \mathbf{g}/\|\mathbf{g}\|_2$

    Compute $\mathbf{h} = (\mathbf{v}^T \dfrac{\partial \mathbf{g}}{\partial \mathbf{u}})^T = \dfrac{\partial \mathbf{f}}{\partial \mathbf{x}}\mathbf{v}$, using reverse mode

    Update $\mathbf{u} = \mathbf{h}/\|\mathbf{h}\|_2$

    Compute parameter update from $\dfrac{\partial}{\partial \theta}\left(\mathcal{L} + \lambda \mathbf{u}^T \mathbf{h}\right)$

**end**

---

|                | ReLU | MaxMin | GroupSort-4 | FullSort | Maxout |
|----------------|------|--------|-------------|----------|--------|
| Standard       | 1.61 | 1.47   | 1.62        | 3.53     | 1.40   |
| Dropout        | 1.27 | 1.37   | 1.29        | 3.62     | 1.27   |
| Björck         | 1.54 | 1.25   | 1.43        | 2.06     | 1.43   |
| Spectral Norm  | 1.54 | 1.26   | 1.32        | 2.94     | 1.26   |
| Spectral Jac   | 1.05 | 1.09   | 1.24        | 1.93     | 1.02   |
| Parseval       | 1.43 | 1.40   | 1.44        | 3.36     | 1.35   |
| $L_\infty$     | 2.25 | 2.28   | 2.22        | 4.88     | 1.98   |

Table 4: **MNIST classification** Test error shown for different architectures and activation functions.

## F  ADDITIONAL EXPERIMENTS

In this section we present additional experimental results which the main paper did not have space to support.

### F.1  CLASSIFICATION

We compared a wide range of Lipschitz architectures and training schemes on some simple benchmark classification tasks. We demonstrate that we are able to learn Lipschitz neural networks which are expressive enough to perform classification without sacrificing performance.

**MNIST Classification**  We explored classification with a 3-layer fully connected network with 1024 hidden units in each layer. Each model was trained with the Adam optimizer (Kingma & Ba, 2014). The full results are presented in Table. 4.

For all models the GroupSort activation is able to perform classification well - especially when the Lipschitz constraint is enforced. Surprisingly, we found that we could even apply the GroupSort activation to sort the entire hidden layer and still achieve reasonable classification performance, even when using dropout. When aiming to train good classifiers we found that spectral Jacobian regularization was most effective (Appendix E).

While the Parseval networks are capable of learning a strict Lipschitz constraint this does not always hold in practice. A small beta value leads to slow convergence towards orthonormal weights. When early stopping is used, which is typically important to achieve good validation accuracy, it is difficult to ensure that the resulting network is indeed 1-Lipschitz.

**Classification with little data**  While enforcing the Lipschitz constraint aggressively could hurt overall predictive performance, it decreases the generalization gap substantially. Motivated by the observations of Bruna & Mallat (2013) we investigated the performance of Lipschitz networks on small amounts of training data, where learning robust features to avoid overfitting is critical.

For these experiments we kept the same network architecture as before. We trained standard unregularized networks, networks with dropout, networks regularized with weight decay, and 1-Lipschitz

| Data Size | Standard | | Dropout | | Weight Decay | | Björck | |
|---|---|---|---|---|---|---|---|---|
| | ReLU | MaxMin | ReLU | MaxMin | ReLU | MaxMin | ReLU | MaxMin |
| 300 | 12.40 | 12.14 | 7.30 | 10.64 | 11.06 | 10.81 | 8.12 | 7.81 |
| 500 | 8.57 | 9.13 | 5.54 | 6.15 | 7.33 | 7.50 | 5.96 | 6.98 |
| 1000 | 5.95 | 6.23 | 3.70 | 4.58 | 5.14 | 6.05 | 4.45 | 4.54 |
| 5000 | 2.54 | 2.51 | 1.84 | 2.15 | 2.31 | 2.55 | 2.23 | 2.31 |
| 10000 | 1.77 | 1.76 | 1.26 | 1.70 | 1.58 | 1.57 | 1.66 | 1.64 |

Table 5: **MNIST Classification with limited training data** Test error for varying architectures and activations per training data size.

| | Standard | | Parseval | | Spec Jac Regularization | |
|---|---|---|---|---|---|---|
| | ReLU | MaxMin | ReLU | MaxMin | ReLU | MaxMin |
| CIFAR-10 | 95.29 | 94.57 | 95.45 | 94.83 | 95.44 | 94.62 |

Table 6: **CIFAR-10 Classification** Test accuracy for Wide ResNets (Depth 28, Width 4) with varying activations and training schemes.

neural networks enforced with the Björck algorithm. In these experiments we are using a LeNet-5 architecture, with convolutions and max-pooling — the latter prevents norm preservation and thus may reduce the effectiveness of MaxMin substantially. We found that Dropout was the most effective regularizer in this case but confirmed that networks with Lipschitz constraints were able to significantly improve performance over unregularized networks. Full results are in Table 5.

**Classification on CIFAR-10**  We briefly explored classification on CIFAR-10 using Wide ResNets (Depth 28, Width 4) (Zagoruyko & Komodakis, 2016; He et al., 2016). We performed these experiments primarily to explore the effectiveness of the MaxMin activation in a more challenging setting. We stuck with the optimal optimization hyperparameters for ReLU with SGD and performed a small search over regularization parameters for Parseval and Spec Jac regularization. We present results in Table 6. We found that MaxMin performed comparably to ReLU in this setting and hope to explore this further in future work.

## F.2 Training WGAN-GP

We found that the MaxMin activation could also be used as a drop-in replacement for ReLU activations in WGAN architectures that utilize a gradient-norm penalty in the training objective. We took an existing implementation of WGAN-GP which used a fully convolutional critic network with 5 layers and LeakyReLU activations. The generator used a linear layer followed by 4 deconvolutional layers. We trained this model with the tuned hyperparameters for the LeakyReLU activation and then used the same settings to train a model with MaxMin acivations. We defer a more thorough study of this setting to future work but present here the output of the trained generators after 50 epochs of training on the CelebA dataset (Liu et al., 2015) in Figure 15.

## F.3 Dynamical Isometry

Gradient norm preservation also enables our methods to represent functions whose input-output Jacobian has singular values that all concentrate near unity (Pennington et al., 2017), a property known as *dynamical isometry*. This property has been shown to speed up training by orders of magnitude when enforced during weight initialization (Pennington et al., 2017; Sokol & Park, 2018), and explored in the contexts of training RNNs (Chen et al., 2018) and very deep convolutional neural networks (Xiao et al., 2018). Enforcing gradient norm preservation on each layer also effectively solves the vanishing gradient problem, as the L2 norm of the back-propagated gradients are maintained at unity throughout the neural network. Using our methods, (Björck Orthonormalization (Björck & Bowie, 1971) and GroupSort), one can maintain dynamical isometry throughout training, reaping the aforementioned benefits. Interestingly, ReLU networks are not capable of achieving dynamical isometry (Pennington et al., 2017).

In Figure 16 we plot the distribution of all singular values of ReLU and GroupSort 2-norm-constrained networks trained as MNIST classifiers. While the ReLU singular values are spread in the range 4-8 the GroupSort network concentrates the singular values in range 9-10. Dynamical

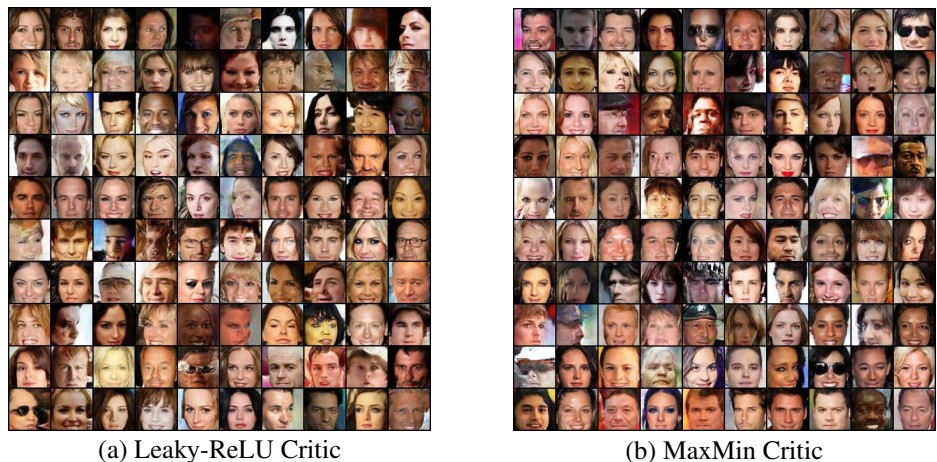

(a) Leaky-ReLU Critic          (b) MaxMin Critic

Figure 15: Generated images from WGAN-GP models trained on the CelebA dataset.

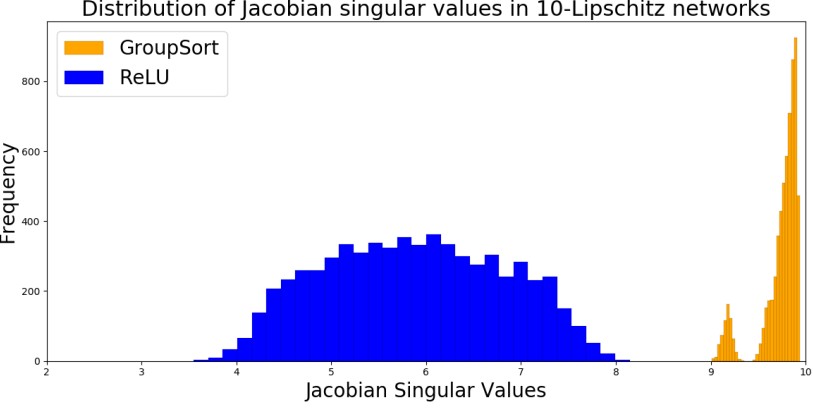

Figure 16: **Jacobian singular values distribution** We compare the Jacobian singular values of ReLU and GroupSort networks.

isometry (Pennington et al., 2017) requires all Jacobian singular values to be concentrated around 1. Typically this property is defined with respect to the initialization of the weights but using 2-norm constraints and GroupSort activations we are able to approximately achieve dynamical isometry throughout training. We leave further investigations into exploiting these benefits on practical problems to a future study.

## G    EXPERIMENT DETAILS

Here we present additional details of the experiments conducted in the main paper.

### G.1    SIMPLE PROBABILITY DISTRIBUTIONS AND THEIR CORRESPONDING DUAL SURFACES

**Absolute value:**    We pick $p_1(\mathrm{x}) = \delta_0(x)$ and $p_2(\mathrm{x}) = \frac{1}{2}\delta_{-1}(x) + \frac{1}{2}\delta_1(x)$, where $\delta_\alpha(x)$ stands for the Dirac delta function located at $\alpha$. It can be shown that the optimal dual surface learned while computing the Wasserstein distance between $p_1$ and $p_2$ is the absolute value function. This also makes intuitive sense, as the function that assigns "as low values as possible" at $x = 0$ and assigns "as low values as possible" at $x = -1$ and $x = 1$ while making sure that the absolute value of the slope of the function never exceeds 1, must be the absolute value function.

The Wasserstein distance obtained using absolute value as the dual function is 1. This becomes clearer when viewed from the primal problem, as the transport plan that will minimize the primal objective will simply be to map the center Dirac delta equally to the ones near it. This requires all the unit masses to be moved by a distance of 1.

The networks we trained had 3 hidden layers each with 128 hidden units.

**Multiple 2D Circular Cones:**    We describe the probability distributions $p_1$ and $p_2$ implicitly by describing how we sample from them.  $p_1$ is sampled from by selecting one of the three points $((-2, 0), (0, 0)$ and $(2, 0))$ uniformly.  $p_1$ is sampled from by first uniformly selecting one of the three points aforementioned, then uniformly selecting a point on the circle surrounding it, with radius 1. Hence Wasserstein dual problem aims to find a Lipschitz function which assigns "as high as possible" values to the three points, and "as low as possible" values to the circles with radius 1 surrounding the three points. Hence, the optimal dual function must consist of three cones centered around $(-2, 0)$, $(0, 0)$ and $(2, 0)$. The behavior of the function outside this support doesn't have an impact on the solution.

The Wasserstein distance between $p_1$ and $p_2$ is equal to 1. From the perspective of the primal formulation, the optimal transport plan must simply consist of mapping the probability mass to the nearby circles surrounding them uniformly. This leads to an expected transport (Wasserstein distance) cost of 1.0.

The networks we trained had 3 hidden layers each with 312 hidden units.

**$n$ Dimensional Circular Cones:**    This is a simple extension of the absolute value case described above.

Here, we check how the performance of architectures built with different activation functions as we increase input dimensionality. We pick $p_1$ as the Dirac delta function located at the origin, and sample from $p_2$ by uniformly selecting a point from high dimensional spherical shell with radius 1, centered at the origin. Following similar arguments developed for absolute value and multiple 2D cones, it can be shown that the optimal dual function is a single high dimensional circular cone and the Wasserstein distance is also equal to unity.

### G.2    WASSERSTEIN DISTANCE ESTIMATION

The GAN variants we trained on MNIST and CIFAR10 datasets used the WGAN formulation first introduced in Arjovsky et al. (2017), and improved by Gulrajani et al. (2017) respectively. The architectures of the generator and critic networks were the same as the ones used in(Chen et al., 2016). For the subsequent task of Wasserstein distance estimation, the weights of the generator networks were frozen after the initial GAN training has converged. For the norm-constrained critics we used a shallow fully connected architecture (3 layers with 720 neurons in hidden each layers).

## G.3 CLASSIFICATION

For the MNIST classification task we search of the hyperparameters are follows. For the Björck, $L_\infty$ constrained, and Spectral Norm architectures we try networks with a guaranteed Lipschitz constant of 0.1, 1, 10 or 100. For Parseval networks we tried $\beta$ values in the range 0.001, 0.01, 0.1, 0.5. For spectral Jacobian regularization we scaled the penalty by 0.01, 0.05, or 0.1.

In order to scale the Lipschitz constant of the network, we introduce constant scaling layers in the network such that the product of the constant scale parameters is equal to the Lipschitz constant. As the activation functions are homogeneous, e.g. $\text{ReLU}(a\mathbf{x}) = a\text{ReLU}(\mathbf{x})$, this is equivalent to scaling the output of the network as described in Section 4.

## G.4 ROBUSTNESS AND INTERPRETABILITY

For the adversarial robustness experiments we trained fully-connected MNIST classifiers with 3 hidden layers each with 1024 units. We used the $L_\infty$ projection algorithm referenced in Section 4.2. We applied the projection to each row in the weight matrices after each gradient update, but found that applying the projection during the forward pass worked equally well and had similar computational overhead.

Our implementation of the FGS attack is standard but we found that the loss proposed by Carlini & Wagner (2016) (in particular, $f_6$ which the authors found most effective) was necessary to generate attacks for the Margin-0.3 MaxMin network (and produced stronger adversarial examples for the other networks). PGD also had difficulty generating adversarial examples for the Margin-0.3 MaxMin network. We found it was necessary to run PGD for 200 iterations and to use a scaled down version of the random initialization typically used: instead of randomly perturbing $x$ in the $\epsilon$ ball we perturbed it by at most $\epsilon/10$ and then ran the usual scheme.

For the intepretable gradients in Figure 7 we used the same architecture, but trained the network with 2-norm projections. We chose a random image from each class (0-4 only) and computed the input-output gradient with respect to the loss function. In the image, We found that similar results were achieved with $\infty$-norm projections (and hinge loss) but the uniform gradient scale made the 2-norm-constrained input-output gradients easier to visualize.

