# OpenReview forum: "Sorting out Lipschitz function approximation"
_ICLR.cc/2019/Conference_

### Official Review · AnonReviewer1 · 2018-10-29
**Potentially interesting but unfinished work**

**Rating:** 4
**Confidence:** 4

**Review:**

The paper proposes a new "sorting" layer in neural networks that offers
some theoretical properties to be able to learn network which are 1-Lipschitz
functions.

The paper contains what seems to be a nice contribution but the manuscript
seems to have been written in a rush which makes it full of typos
and very hard to read. This unfortunately really feels like unfinished work.

Just to name a few:

- Please check the use of \citep and \citet. See eg Szegedy ref on page 3.

- Unfinished sentence "In this work ..." page 3.

- "]" somewhere at the bottom of page 4.

- "Hence, neural network has cannot to lose Jacobian norm... " ???

etc...

Although I would like to offer here a comprehensive review I consider
that the authors have not done their job with this submission.

---

> ### Author Response · Authors · 2018-11-13
> **Sorry for the poor presentation. Please, take another look!**
>
> We are deeply sorry that you felt the paper was not in a position to be given a complete review. We acknowledge that the paper was certainly lacking polish (as also noted by reviewer 2) and accept that this may have made the paper difficult to read in places.
>
> We have uploaded a revised version which is tidier and without so many of the unfortunate errors you spotted previously. The revised version also presents the theoretical results more cleanly with some substantial improvements to the experiments. We hope that you will provide a more complete review at this time.

---

### Official Review · AnonReviewer2 · 2018-11-02
**Interesting paper but missing details and some formal polishing required**

**Rating:** 5
**Confidence:** 4

**Review:**

summary:

A paper that states that a new activation function, which sorts coordinates in a vector by groups, is better than ReLU for the approximation of Lipschtiz functions.

pros:

- interesting experiments
- lots of different problems evaluated with the technique

cons:

- the GroupSort activation is justified from the angle of approximating Lipschitz transformations. While references are given why Lip is good for generalisation, I cannot see why GroupSort does not go *against* the ability of deep architectures to integrate the topology of inputs (see below).
- the proof of Theorem 1 requires polishing (see below)
- experiments require some polishing

detail:

* The proof of Theorem 1 has three problems, first in the main file argument: since ReLU is not differentiable, you cannot use the partial derivative. Maybe a sub differential ? Second, in the RHS after the use of the Cauchy-Schwartz inequality (no equation numbering…) you claim that the product of all three norms larger than 1 implies *each* of the last two is 1. This is wrong: it tell nothing about the the value of each, only about the *product* of each, which then make the next two identities a sufficient *but not necessary* condition for this to happen and invalidates the last identity. Last, the Theorem uses a three lines appendix result (C) which is absolutely not understandable. Push this in the proof, make it clear.

Section D.1 (proof of Theorem 2) the proof uses group size 2 over a vector of dimension 2. This, unless I am mistaken, is the only place where the group sort activation is used and so the only place where GroupSort can be formally advocated against ReLU. If so, what about just using ReLUs and a single group sort layer somewhere instead of all group sort ? Have the authors tried this experimentally ?

If I strictly follow Algorithm 1, then GroupSort is carried out by *partitioning* the [d] indexes in g groups of the same size. This looks quite arbitrary and for me is susceptible to impair the capacity of deep architectures to progressively integrate the topology of inputs to generalise well. Table 3 tends to display that this is indeed the case as FullSort does much worse than ReLU.

* Table 5: replace accuracies by errors, to be consistent with other tables.

* in the experiments, you do not always specify the number of groups (Table 4)

---

> ### Author Response · Authors · 2018-11-13
> **Feedback integrated into revised version. Thank you!**
>
> Thank you for your detailed comments. We have uploaded a revised version of the paper which we believe addresses the majority of your concerns. You can find more detailed responses below.
>
> Concern 1: Is GroupSort leading to bad networks? (Integrate the topology of inputs)
>
> Could you clarify what you mean by “integrate the topology of inputs”? Interpreting this as “is GroupSort a niche activation?”, we respond with the following: GroupSort is able to recover many common activation functions, for example ReLU, MaxOut, Concatenated ReLU, absolute value (now detailed in Appendix A). Importantly, it is often able to do this even with norm constrained weights (note that ReLU cannot recover GroupSort in this case). The main difference then will be how difficult GroupSort networks are to train. We have found practically that GroupSort networks are typically as easy to train as their ReLU counterparts. We trained wide ResNets using MaxMin and achieved comparable performance to ReLU. We also trained CelebA WGANs using MaxMin activations in the critic network without any issues. Importantly, in each case we used the suggested optimization hyperparameters tuned for ReLU and found that MaxMin worked too.
>
> Concern 2: Proof of Theorem 1 is incorrect.
>
> Thank you for taking the time to carefully investigate this result. While we are confident that the result is correct, we have rewritten the proof in an attempt to make it clearer.
>
> To address your points directly: we have modified the statement to hold almost everywhere, in which case we need not discuss sub differentials and may use differentiability directly. For your comment about the Cauchy-Schwarz inequality, note that the product cannot be larger than 1, as each individual component of the product has to be less than or equal to 1 (by the 1-Lipschitz constraint). Hence, each component must itself be 1. We have made this explicit in the revised proof, by bounding the product of norms above and below by 1. We have also removed the three-line result expressed in the appendix and instead baked it into the proof as part of the induction step. Finally, we have extended this result to hold in the setting of vector-valued inputs. Thank you for pointing out these issues to us. We hope that the improvements we’ve presented will clarify the proof for you but would be happy to discuss this further.
>
> Concern 3: Why not use GroupSort only at the end of the network?
>
> The universal construction must use GroupSort for the intermediate layers as well. We construct the final network by taking the max/min of increasingly wide and deep networks  (which are themselves max/mins). The final result is a network which uses MaxMin throughout and is able to represent the max/min of arbitrarily complicated Lipschitz functions.
>
> Concern 4: Table 3 shows FullSort doing worse
>
> This is true and perhaps not particularly surprising. In Section 4.1 (Section 3.2 in old version) we state that while FullSort and MaxMin are equally expressive, the former leads to a more challenging optimization problem. The full-sort activation sorts the entire activation vector. We were surprised that the network was able to learn anything reasonable at all (especially with dropout!) and presented this column as a surprising observation - we do not suggest that practitioners adopt FullSort for classification as it is harder to optimize and more computationally expensive. We would be happy to clarify this further in the paper.
>
> We hope that our responses above adequately resolve your concerns. Although we believe the current revision does a much better job of presenting these arguments, we warmly encourage you to provide any criticisms that may help us further express these points more clearly.

---

### Official Review · AnonReviewer3 · 2018-11-05
**Review of "Universal Lipschitz Functions"**

**Rating:** 7
**Confidence:** 3

**Review:**

This paper introduces GroupSort. The motivation is to find a good way to impose Lipschitz constraint to the learning of neural networks. An easy approach is "atomic construction", which imposes a norm constraint to the weight matrix of every network layer. Although it guarantees the network to be a Lipschitz function, not all Lipschitz functions are representable under this strong constraint. The authors point out that this is because the activation function of the network doesn't satisfy the so called Jacobian norm preserving property.

Then the paper proposes the GroupSort activation which satisfies the Jacobian norm preserving property. With this activation, it shows that the network is not only Lipschitz, but is also a universal Lipschitz approximator. This is a very nice theoretical result. To my knowledge, it is the first algorithm for learning a universal Lipschitz function under the architecture of neural network. The Wasserstein distance estimation experiment confirms the theory. The GroupSort network has stronger representation power than the other networks with traditional activation functions.

Admittedly I didn't check the correctness of the proof, but the theoretical argument seems like making sense.

Despite the strong theoretical result, it is a little disappointing to see that the GroupSort doesn't exhibit any significant advantage over traditional activation function on image classification and adversarial learning. This is not surprising though.

---

> ### Author Response · Authors · 2018-11-13
> **Thank you for the feedback**
>
> Thank you for your kind feedback!
>
> We agree with your comments on the empirical results presented in the original paper. We are pleased to present several improvements in our revised version. We include much improved adversarial robustness results which contain provable robustness guarantees and strong empirical evidence that MaxMin leads to significantly more expressive networks than ReLU (see Fig. 8 in revised version). We also compared MaxMin to ReLU on Wide ResNets and found that MaxMin had comparable performance over the training schemes we explored (we used a limited hyperparameter search around the optimal ReLU settings). Finally, we used MaxMin to train a WGAN-GP model on CelebA and generated images qualitatively on-par with the carefully tuned Leaky-ReLU model. We believe that these new additions show that MaxMin is more than just a niche activation function and in Lipschitz-constrained settings may lead to significant practical gains.

---

### Public Comment · (anonymous) · 2018-11-06
**Related work: OPLU (orthogonal permutation linear unit)**

MaxMin, GroupSort with a grouping size of 2, looks the same as OPLU (orthogonal permutation linear unit) proposed in https://arxiv.org/abs/1604.02313. The motivation of OPLU was also norm preserving.

---

> ### Author Response · Authors · 2018-11-06
> **Thank you for bringing this to our attention**
>
> Thank you for bringing this preprint to our attention! OPLU is indeed identical to GroupSort with a grouping size of 2 (which we call MaxMin). We will cite this paper and credit it for proposing MaxMin and observing that it is norm-preserving.
>
> The focus of the OPLU paper is to preserve the norm of gradients during backpropagation to allow the training of extremely deep networks. In our latest revision of the paper, we also discuss this property in terms of dynamical isometry [1]. In our work, our primary focus is on training expressive Lipschitz-constrained architectures and we identify gradient norm preservation as an important condition for which MaxMin is one such suitable activation. We also prove that using MaxMin we are able to recover universal approximation of Lipschitz functions (which other common activations fail to achieve).
>
> The revised version and our response to the reviewers will be posted soon.
>
> [1]: Pennington et al. “Resurrecting the sigmoid in deep learning through dynamical isometry: theory and practice” https://arxiv.org/abs/1711.04735

---

### Author Response · Authors · 2018-11-13
**Uploaded revision and individual comments**

We thank each of the reviewers for their time and comments. We have uploaded a revised version of our paper which addresses the notes from each reviewer and includes substantial improvements to the writing. The new version provides improved presentation of theoretical content and some new additions to the experiments section. We emphasize that the scope of the paper has not changed at all. Alongside these changes, we have also modified the title of our paper to “Sorting out Lipschitz function approximation”.

We have also responded to each of the reviewers in kind and welcome further discussion!

---

### Meta-Review · Area_Chair1 · 2018-12-15
**interesting and potentially impactful idea but needs revisions**

**Confidence:** 4
**Recommendation:** Reject

**Metareview:**

This paper presents an interesting and theoretically motivated approach to imposing Lipschitz constraints on functions learned by neural networks. R2 and R3 found the idea interesting, but R1 and R2 both point out several issues with the submitted version, including some problems with the proof--probably fixable--as well as a number of writing issues. The authors submitted a cleaned-up revised version, but upon checking revisions it appears the paper was almost completely re-written after the deadline. I do not think reviewers should be expected to comment a second time on such large changes, so I am okay with R1's decision to not review the updated version. Future reviewers of a more polished version of the paper will be in a better position to assess its merits in detail.

---

> ### Author Response · Authors · 2018-12-22
> **\citet vs. \citep is no excuse not to write a real review**
>
> We understand that R1 is probably very busy and did not want to read our paper twice. But we would have appreciated if they could find the time to read it once.